# Patient Perception When Transitioning from Classic to Remote Assisted Cardiac Rehabilitation

**DOI:** 10.3390/diagnostics12040926

**Published:** 2022-04-07

**Authors:** Ștefan-Sebastian Busnatu, Maria-Alexandra Pană, Andreea Elena Lăcraru, Cosmina-Elena Jercălău, Nicolae Paun, Massimo Caprino, Kai Gand, Hannes Schlieter, Sofoklis Kyriazakos, Cătălina Liliana Andrei, Crina-Julieta Sinescu

**Affiliations:** 1Department of Cardiology, University of Medicine and Pharmacy “Carol Davila”, Emergency Hospital “Bagdasar-Arseni”, 050474 Bucharest, Romania; stefan.busnatu@umfcd.ro (Ș.-S.B.); andreea.lacraru@umfcd.ro (A.E.L.); cosmina-elena.jercalau@rez.umfcd.ro (C.-E.J.); catalina.andrei@umfcd.ro (C.L.A.); crina.sinescu@umfcd.ro (C.-J.S.); 2Department of Cardiology, University of Medicine and Pharmacy “Carol Davila”, Clinical Hospital “Theodor Burghele”, 020021 Bucharest, Romania; nicolae.paun@umfcd.ro; 3Department of Neurorehabilitation Sciences, Casa Cura Policlinico, 20144 Milano, Italy; m.caprino@ccppdezza.it; 4Research Group Digital Health, Faculty of Business and Economics, Technische Universität Dresden, 01062 Dresden, Germany; kai.gand@tu-dresden.de (K.G.); hannes.schlieter@tu-dresden.de (H.S.); 5Department of Business Development and Technology, Aarhus University, 7400 Aarhus, Denmark; sofoklis@btech.au.dk

**Keywords:** virtual assistant, cardiac rehabilitation, machine learning, telerehabilitation

## Abstract

Cardiac rehabilitation is an individualized outpatient program of physical exercises and medical education designed to accelerate recovery and improve health status in heart disease patients. In this study, we aimed for assessment of patients’ perception of the involvement of technology and remote monitoring devices in cardiac recovery. During the Living Lab Phase of the Virtual Coaching Activities for Rehabilitation in Elderly (vCare) project, we evaluated eleven patients (five heart failure patients and six ischemic heart disease patients). Patient admission in the UMFCD cardiology clinical department served as a shared inclusion criterion for both study groups. In addition, the presence of II or III heart failure NYHA stage status was considered an inclusion criterion for the heart failure study group and patients diagnosed with ischemic heart disease for the second one. We conducted a system usability survey to assess the patients’ perception of the system’s technical and medical functions. The survey had excellent preliminary results in the heart failure study group and good results in the ischemic heart disease group. The limited access of patients to cardiac rehabilitation in Romania has led to increased interest and motivation in this study. The final version of the product is designed to adapt to patient needs and necessities; therefore, patient perception is necessary.

## 1. Introduction

Cardiac rehabilitation is an individualized outpatient program of physical exercises and medical education designed to accelerate recovery and improve health status in heart disease patients [1]. 

A cardiac rehabilitation program is structured with three stages: the acute phase (in hospital), the subacute phase (center-based), and outpatient therapy (home-based). Unfortunately, the second and third phases of a cardiac rehabilitation program face many barriers of implementation with an overall participation of under 50%, despite international guidelines prescribing recommendations [2,3]. 

The main factors leading to low adherence of patients to cardiac rehabilitation can be classified into person-related (advanced age, female gender, unemployed status, low education, and low-income background) and aspect-related (rural areas, lack of means of transport) [4]. 

### 1.1. Cardiovascular Disease: Epidemiology, Economic Impact, and Management Strategies

The 21st century is characterized by a longer life expectancy as the number of people aged over 65 years is increasing. It is expected that by 2030 the percentage of people over 65 years old will grow from 17.4% to 25.6%. The ageing of the population translates into a high prevalence of cardiac diseases and loss of cognitive and physical autonomy. It represents a medical and social issue, which should be addressed sooner rather than later in order to secure the future well-being of this fragile group of patients [5].

Heart disease has been the leading cause of death worldwide for the past 20 years, and the number is steadily rising, from 2 million deaths per year in 2000 to nearly 9 million deaths in 2019 [6]. 

Clinical and epidemiological studies developed short- and long-term risk prediction algorithms for identifying patients at high-risk of developing cardiovascular disease. These patients are the main beneficiaries of reducing cardiovascular risk factors through primary prevention programs [7].

Globally, the costs of hospitalization, drug or interventional treatment, and monitoring visits of cardiovascular disease patients are huge and increasing alongside patient life expectancy. As an example, in the United States of America in 2010, medical costs were approximately USD 863 billion, with an estimated increase of up to USD 1 trillion by the year 2030 [8,9]. 

Home-based cardiac rehabilitation is a way for patients to enjoy all the benefits of clinician-supervised cardiac recovery from the comfort of their homes, under medical supervision. In addition, virtual communication, the development of digital modules, and the monitoring of vital signs are all possible with wearable devices [10]. 

Although it represented an alternative for many years, there were no premises of the rapid implementation of online cardiac rehabilitation. The COVID-19 pandemic raised the opportunity for rapid integration among the services offered by healthcare providers. These services were translated into telemedicine and the use of remote assistance devices [11]. 

The switch from classic to online cardiac rehabilitation means waiting time reduction, flexible participation schedules, no need for the patient to travel, and benefits in terms of morbidity, mortality, and quality of life [12,13].

Beyond the COVID-19 pandemic, adopting technology as the standard practice looks promising and might be the best alternative for specific cardiac patients in terms of efficiency in secondary prevention management of cardiovascular disease [14]. 

### 1.2. Virtual Assistants: A Solution to Telerehabilitation Implementation Difficulties in Eastern European Countries

Although ambulatory cardiac rehabilitation programs prove to be efficient, they are rarely prescribed to patients due to uneven access to health services [15].

Telerehabilitation could be the answer to this problem. Current literature supports the use of digital technology to address health inequalities and create compelling and constant changes in medical services offered. The ability of telerehabilitation to overcome temporal and spatial barriers is highlighted even more in the current COVID-19 pandemic situation [16].

Romania and many other European Countries face health inequalities, as significant numbers of patients are not admitted to rehabilitation programs, mostly due to lack of financial resources and lack of reimbursement. A recent study investigating key barriers in cardiac rehabilitation in East-Central Europe highlighted demographic issues as one of the most significant drawbacks for rehabilitation adherence [17,18]. 

In a successful implementation, rural communities would be the main beneficiaries of these services. Despite these limitations, the overall image is auspicious. Digital health technology could answer many of the challenges faced by on-site cardiac rehabilitation. Virtual coaches, also known as e-coaches, can vary, from classic rule-based smartphone apps that guide the patient, to more advanced machine learning algorithms, that intelligently support patient rehabilitation with automatic adaptation of clinical pathways. Virtual coaches can provide medical follow-up in the community and can reach out to demographically remote areas that are difficult to access by health care providers [19,20].

Apart from improving patient quality of life, the virtual coach optimizes the economics of medical and social care treatments [21]. 

One example of a virtual assistant under development is the virtual coach for rehabilitation in the elderly patients developed in the vCare project. Under the funding of the European Union’s Horizon 2020 Research and Innovation Programme grant no 769807, the vCARE project has the overall objective to support the recovery of active and independent life at home [22]. 

The development of the vCARE project involves twelve partners from seven European countries with a multidisciplinary consortium of researchers, healthcare providers, and industry experts. The virtual coach must be used in different national systems for various pathologies and rehabilitation settings to ensure a multi-site and multi-setting approach [22].

The chosen pathologies are stroke, Parkinson’s disease, heart failure, and ischemic heart disease, divided among the healthcare providers participating in the project [22].

One of the project sub-goals is evaluating patient perception of the virtual assistant as a constant in their lives, an essential parameter for patient’s motivation and further participation in the system testing, and a vital evaluation of the public receptivity of virtual coaches [22]. 

Recent article reviews of patient perception of telemedicine state a 95–100% patient satisfaction rate compared with classic medical appointments [23].

Decreased time travel, the presence of administrative support, reliable and easy-to-use technology, and adequate reimbursement of medical services are a few of the positive arguments patients cite in favor of telemedicine [23]. 

In the center of the active development of telemedicine and remote medical assistance, one should see patient perception as essential for technology’s future growth and implementation [24,25]. 

Telemedicine represents a healthy mix of medical care and technology. Therefore, it should be instituted in all urban medical centers to support rural, isolated areas, acting as a bridge for broader accessibility of patients to medical services and a good strategy in the second prevention management of cardiovascular disease and more [26]. 

### 1.3. Study Goals

The main purpose of this study was to assess patient perceptions of virtual assistant use in the process of remote cardiac recovery. Given the novelty of the technology, we simultaneously analyzed the profile of the patient who would benefit from this service, the degree of technology used at home, and the desire in accessing such service. We consider all these key parameters when evaluating the opportunity to launch a product on the medical market.

## 2. Materials and Methods

### 2.1. Participants and Procedures

In the active development of innovative healthcare technology, both medical staff and patients should evaluate its acceptability and usability throughout the entire project.

The Living Lab Phase of the vCARE project was oriented towards three significant parameters: utility, efficacy, and adaptability, to be able to quantify the adoptability of the virtual solution. This phase’s primary goal was to validate the functionality of the digital platform, its interactions with patients, and its adaptive capacities according to individual patient opinion. 

The Heart Failure (HF) and Ischemic Heart Disease (IHD) Living Lab Phase took place at the Department of Cardiology of “Carol Davila” University of Medicine and Pharmacy (UMFCD) in Bucharest, Romania for a period of 11 months.

The Living Lab field of study consisted in testing the physical therapy and risk factor modification pathways pre-established by the healthcare providers according to the needs and necessities of the patients, using the virtual assistant, with constant monitoring of specific key performance indicators (Figure 1 and Figure 2).

Five patients were enrolled in the heart failure study group and six in the ischemic heart disease study group. Patient admission in the UMFCD cardiology clinical department served as a shared inclusion criterion. In addition, the presence of a II or III Heart Failure New York Heart Association Functional Classification (NYHA) Stage status was considered an inclusion criterion for the heart failure study group. We selected the patients with stable or unstable angina pectoris, myocardial infarction, ischemic heart disease with medical therapy, history of percutaneous coronary intervention, or coronary bypass surgery for inclusion in the ischemic heart disease study group. The exclusion criteria were age (which had to be over 18 years old) and patients with movement disabilities.

Patient participation in the study was voluntary. All patients received information about the study beforehand. We could not collect patient data completely anonymously, so the pseudo anonymization alternative was chosen. Every participant in the study received an identifier, which separated the personal information from the study’s data collection. Every participant was informed about their right to privacy and the private storage and use of their data. The study conducted did not present any potential physical, social, psychological, or legal harm to patients. Every patient signed to confirm informed consent.

### 2.2. Measures

The initial assessment of patients consisted of an analysis of demographic indices, cardiovascular risk factors, personal medical history, current cardiovascular health status, and level of home technology use.

Each patient in the two study groups followed a cardiac rehabilitation program with 2–3 times a week, for a total of 6 weeks. During each session, they performed specific pathways of physical therapy and risk factor modification using virtual assistance with all wearable devices at their disposal: a blood pressure monitor, smart bracelet, environmental sensors, body-tracking sensors, and an intelligent weight scale. At the end of each patient’s program, we conducted a system usability survey to assess the patient’s perception of the system’s technical and medical functions (Figure 3). 

The System Usability Scale (SUS) is a measuring tool initially created in 1986, which allows users to evaluate a wide variety of technological products and services. Already an industry-standard tool, the SUS is an efficient and easy-to-use scale, reliable for small sample sizes, and is able to differentiate a usable system from an unusable one.

The scale has a 10-item design; each participant scores every item with one of five responses from “strongly agree” to “strongly disagree”. Then, the responses are converted to numbers, added, and multiplied by 2.5 to reach the final scores of 0–40 to 0–100 [27]. 

Relying on research, SUS scores over 68 are above average, and anything less than 68 is below average. However, the correct way of interpreting the results would be to normalize the score to produce a percentile ranking [27]. 

### 2.3. Statistical Analysis

The HF study group had five patients (four men and one woman), and the IHD group had six patients (five men and one woman). The small sample size of our study group was due to the COVID-19 pandemic, and also to the Living Lab pilot test conditions. It was impossible to perform cardiac rehabilitation in safe conditions with more than one patient per session. The primary purpose of this phase was to test the product and effectively integrate all components. The number of patients involved in this stage was desired to be low, precisely because of the possible errors that could occur with the first use of the system. Given the excellent results obtained, it was decided to analyze them, considering the possibility of conforming or refuting them in the subsequent phases of the project currently underway.

Microsoft Excel was used to collect and analyze demographic data, cardiovascular risk factors, and the use of home technology. For the system usability evaluation, the SUS results were entered into an SPSS database for analysis, with a representative creation of the results in the form of box plot figures.

We believe the results obtained are relevant despite the small patient study groups, considering the measuring tool used. The SUS is a reliable tool designed to be used for small sample sizes, and to be able to differentiate a usable system from an unusable one. 

## 3. Results

### 3.1. Sample Description

Study participants’ demographic characteristics and cardiovascular risk factors are presented in Table 1 and Table 2. We observed a prevalence of male patients in both study groups, with a minimum age of 41 years old in the HF group and 24 years old in the IHD group. The maximum age of the patients in the HF group was 72 years old, while in the IHD group, it was 56 years old. More smokers and patients with dyslipidemia were in the IHD group than in the HF group (3:1, respectively, 4:1). On the other hand, all Heart Failure patients were known to have varying degrees of hypertension. At the same time, in the IHD group, there were only three hypertensive patients, of which two had grade one arterial hypertension. In both study groups, patients were either overweight or obese, with a physical activity status of sedentary or low active.

### 3.2. Home Technology Assessment

At study initiation, an evaluation of the technology used at home by the enrolled patients was also performed. Ten items were evaluated, nine of which can be found in Table 3 and Table 4. The tenth item not found in the table was a free-to-answer question about how the patients perceived the technology they were already using. Out of eleven patients, only one heart failure patient did not have an internet connection at home. All patients from both groups were owners of a smartphone, over which they had complete control. All IHD patients and three out of five HF patients had a smart TV. All patients had either a tablet, a laptop, or a computer at home, which they used daily. Only five out of eleven patients used a smart bracelet for pulse monitoring and also used a blood pressure monitor. Six out of eleven patients used an intelligent weight scale at home. Regarding the tenth item of the evaluation form, all patients had a positive answer about the impact of current technology in their lives. They were grateful for the possibility of keeping in touch with their loved ones through video conferencing, for the option of watching their favorite shows without depending on a fixed schedule and for the amount of free information they were able to read at only one click. Regarding the difficulty of using the technology, some patients considered it easy while others found it difficult. Despite the difficulties encountered, they did not give up the use of technology.

### 3.3. System Usability Assessment

The System Usability Survey for the Heart Failure study group had excellent preliminary results: five out of five patients reached good or excellent scores (Figure 4).

The results were weaker in the Ischemic Heart Disease group, with only three out of six patients reaching good or excellent scores. However, the results were over 63, which means that they exceeded the average (Figure 4).

Along with the significant results obtained from evaluating patients’ perceptions, their interest and motivation were the key elements that crowned the success of the Living Lab Phase of the vCARE project. Some of their objective opinions can be found in Figure 5.

## 4. Discussion

One of the key parameters in the successful implementation of medical technology is the perception of the user, who in our case, is the patient. Not only is technology research expensive, but so is its implementation, especially in countries with poor infrastructure. The unfortunate context of the new coronavirus pandemic we are currently experiencing has shown us the importance of technology as a key medical resource, able to streamline the work of healthcare professionals, reduce costs, and expand access to health care in less-favored areas [28].

In addition to testing the product we are currently developing, we had as a sub-goal, the evaluation of patients’ perceptions of the long-term use of technology in cardiac recovery, a medical area as disadvantaged as it is essential in the secondary prevention of cardiovascular disease.

One of the most important findings we encountered is the cardiac patients’ motivations and interest in participating in the study when given the opportunity of cardiac rehabilitation. We believe the reason for this positive attitude is the low availability of cardiac rehabilitation programs in Romania. 

In a recent study, Nabutovsky I. et al., 2020, evaluated through a cross-sectional study, the Israeli cardiovascular patients perception, attitude, and behavioral intention toward remote digital cardiac rehabilitation. The study concluded with positive results, so that more than 80% of the participants were interested in telerehabilitation services if they had been made available to them. These results were not influenced by demographic factors such as age, level of technology used, or gender [29].

Although pleased with our study results, they are only preliminary; we need further studies to certify these outcomes. At first, we thought that the favorable results were based on the young age of the enrolled patients; we therefore analyzed the possibility that the scores could differ in older patients. The SUS results in the ischemic heart disease group slightly contradicted this theory because the lower mean age was not associated with better results. This fact indicates that no matter the age, even little knowledge of technology can come in handy when interacting with a virtual coach. We correlated our system’s usability scale results with the few other results available in the literature. Joao Balsa, Isa Felix et al., 2020, conducted a similar study of an intelligent virtual assistant promoting behavior change and self-care in older people with type 2 diabetes; they obtained an aggregated mean SUS score of 73.75, corresponding to borderline excellent [30].

Haggerty T., Brabson L. et al., 2021, used the System Usability Scale in a two-cycle approach for testing a weight management tool, mWRAPPED. They obtained the first cycle average patient score of 76.5 and an after-revisal average score of 80.5. The study concluded the initial usability of the obesity tool for primary care in patients [31].

The evaluation of the opinion of both medical staff and patients is relevant in many studies. Maciej Banach, Dan Gaita et al., 2020, evaluated medical staff opinion through anonymous surveys in order to highlight implementation barriers of the ADA/EASD (American Diabetes Association/European Association for the Study of Diabetes) guidelines and the optimal measures to overcome them [32]. 

A pilot study conducted by Brewer C. et al., 2017, assessed the virtual word technology of the feasibility and acceptability of cardiac rehabilitation programs. The small study group (8 patients, 25% women) were unanimously satisfied by the method novelty, accessibility, and social connectivity. They reported important improvement in health knowledge and habits secondary to following remote cardiac rehabilitation [33].

Acceptability among patients also emerges from the study results of Banner D. et al., 2015, which strongly support home implementation of virtual cardiac rehabilitation programs for patients recovering from a cardiac event. They highlight benefits such as increased accessibility in disadvantaged environments and a cost-effective method of secondary prevention of cardiovascular disease. As a future certification strategy of cardiac rehabilitation long-term benefits, authors consider mandatory the multiple environmental evaluation (from developed to underdeveloped countries) on more medical complex patient groups [34].

According to our results and those valid in the current literature, the perception evaluated in enrolled patients is statistically significant. The fact that the cardiovascular patient is eager to use technology as a mean of remote cardiac rehabilitation makes development in this area both medically and economically profitable in the future. Given the unfavorable epidemiological context of cardiovascular disease, there should be found effective and inexpensive ways to provide patients with adequate secondary prevention therapies and to not further suffocate the overburdened medical system in the future.

## 5. Conclusions

The virtual assistant we are currently developing has the underlying base of a machine learning approach. The final version of the product must adapt to patient needs and necessities to provide efficient secondary prevention management of their cardiovascular disease. Imagined as the locum tenens of cardiac rehabilitation medical staff, vCARE is the human appearance of post-discharge continued medical care for patients. 

The following steps for the vCARE project are to enroll more patients to test all system functionalities, such as pharmacological intervention, and emotional and social rehabilitation. After the Living Lab, the Pilot Test phase initiates with moving the intelligent system to patient home environments. Finally, we shall compare these collected data with a similar group of patients following conventional cardiac rehabilitation. 

## 6. Limitations 

We identified several limitations to our results. First, the study sample was small, with a prevalence of male patients, so our population might not fully characterize the average Romanian patient. However, the evaluated patient sample provided efficient feedback with a usable system result. Additionally, the presence of the medical staff might have facilitated, in some cases, the use of the system by the patients, further testing being needed in the home environment to certify the actual capacity of the patient in using the techniques and to reevaluate their perception.

## Figures and Tables

**Figure 1 diagnostics-12-00926-f001:**
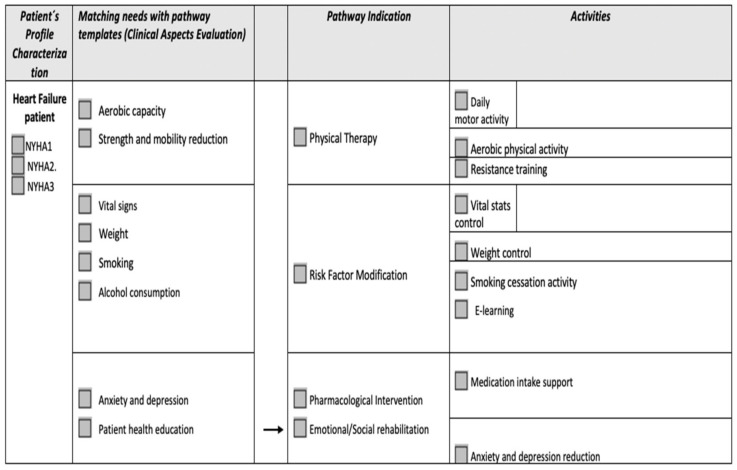
Available pathways for heart failure patient.

**Figure 2 diagnostics-12-00926-f002:**
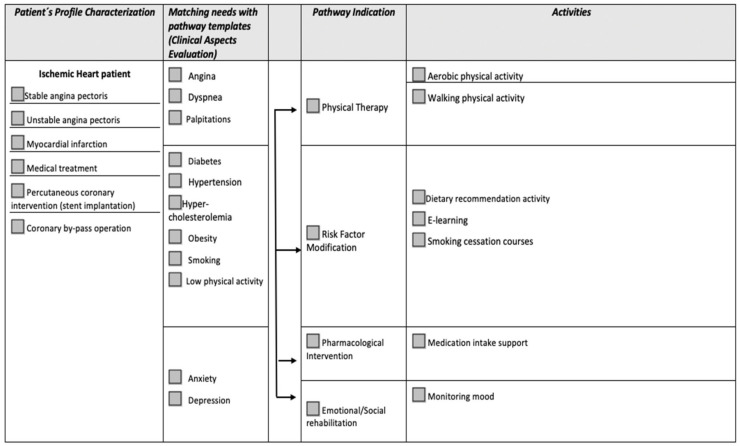
Available pathways for ischemic heart disease patient.

**Figure 3 diagnostics-12-00926-f003:**
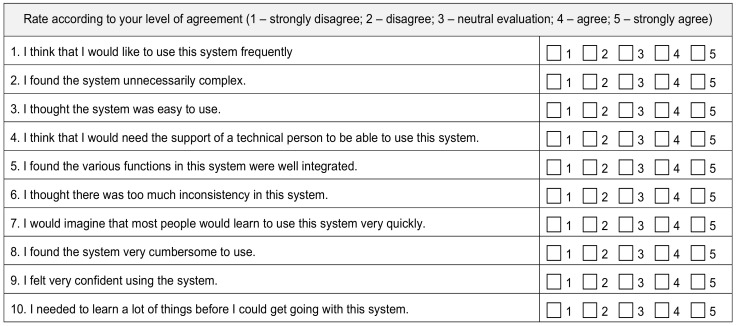
System Usability Survey (SUS).

**Figure 4 diagnostics-12-00926-f004:**
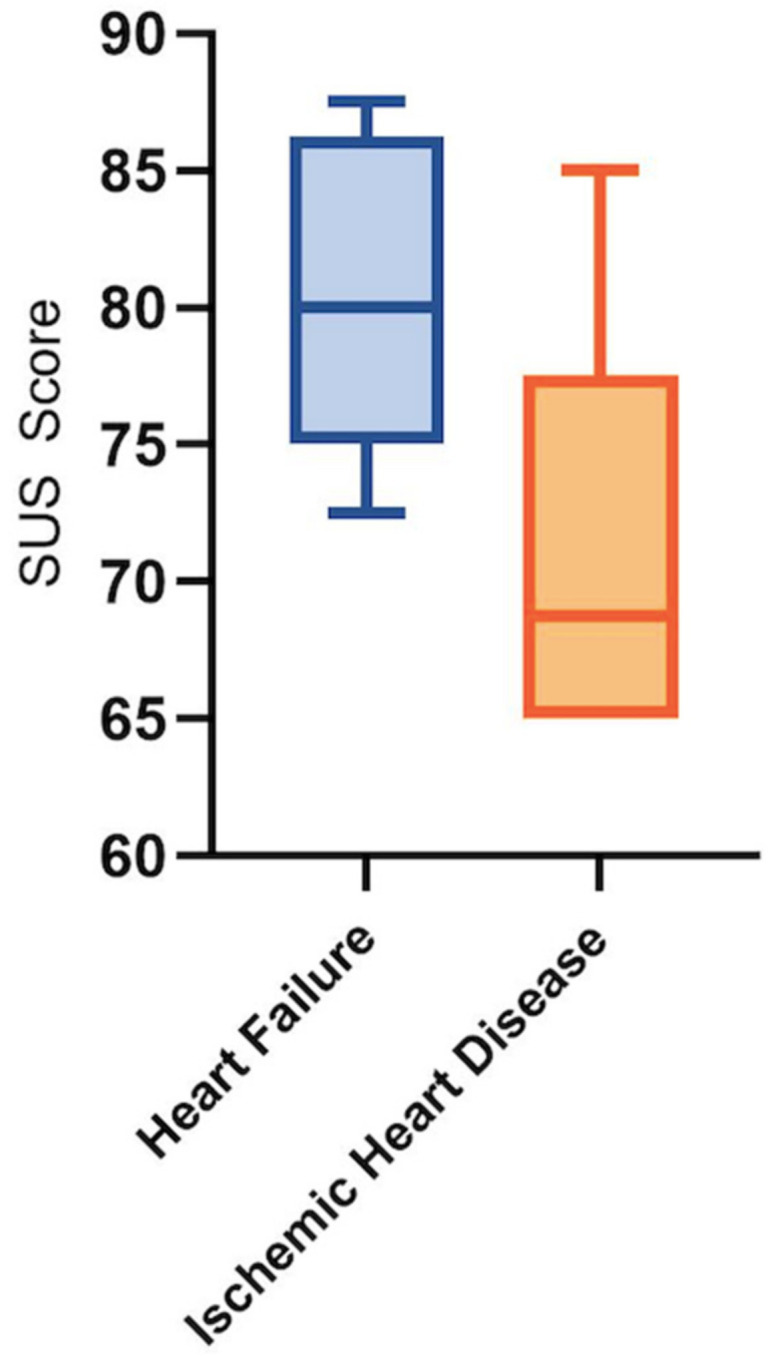
Preliminary results of the System Usability Survey in HF and IHD patients.

**Figure 5 diagnostics-12-00926-f005:**
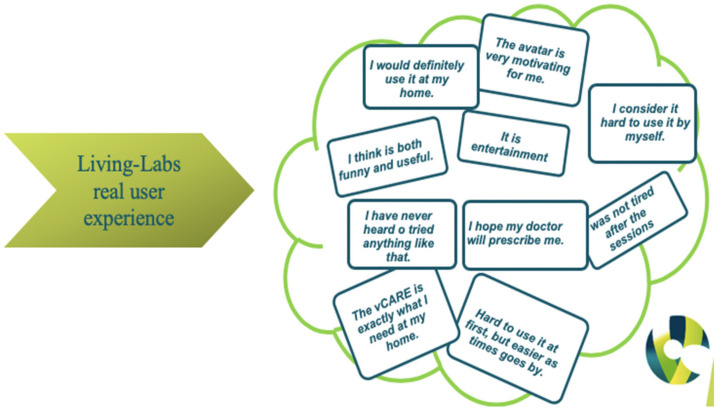
Living-Labs real user experience.

**Table 1 diagnostics-12-00926-t001:** Demographic characteristics and cardiovascular risk factors for HF patients.

Patient	Age	Sex	Smoker Status	AHT	Dyslipidemia	BMI	DM	PA Status
HF_P01	66	M	No	Yes (grade III)	Yes	25.7 kg/m^2^	No	Sedentary
HF_P02	46	M	No	Yes (grade II)	Yes	32.4 kg/m^2^	No	Active
HF_P03	72	F	No	Yes (grade II)	Yes	31.7 kg/m^2^	Yes	Sedentary
HF_P04	65	M	No	Yes (grade III)	No	28.5 kg/m^2^	No	Low active
HF_P05	41	M	Yes	Yes (grade I)	No	34 kg/m^2^	No	Low active

HF = heart failure, AHT = arterial hypertension, BMI = body mass index, DM = diabetes mellitus, PA = physical activity.

**Table 2 diagnostics-12-00926-t002:** Demographic characteristics and cardiovascular risk factors for IHD patients.

Patient	Age	Sex	Smoker Status	AHT	Dyslipidemia	BMI	DM	PA Status
IHD_P01	41	M	No	Yes (grade I)	No	30.4 kg/m^2^	Yes	Low active
IHD_P02	49	M	Yes	No	Yes	28.7 kg/m^2^	No	Low active
IHD_P03	48	F	No	Yes (grade III)	Yes	33.2 kg/m^2^	No	Sedentary
IHD_P04	56	M	No	No	Yes	27.8 kg/m^2^	No	Low active
IHD_P05	24	M	Yes	No	Yes	26.2 kg/m^2^	No	Sedentary
IHD_P06	34	M	Yes	Yes (grade I)	No	29 kg/m^2^	No	Very active

IHD = ischemic heart disease, AHT = arterial hypertension, BMI = body mass index, DM = diabetes mellitus, PA = physical activity.

**Table 3 diagnostics-12-00926-t003:** Home technology HF patients.

Patient	Internet	Smart Phone	Smart TV	Tablet	Laptop	PC	Intelligent Bracelet	BP Monitor	Intelligent Weight Scale
HF_P01	✓	✓				✓	✓	✓	
HF_P02	✓	✓	✓	✓	✓				✓
HF_P03	✓	✓	✓			✓		✓	
HF_P04		✓				✓	✓	✓	
HF_P05	✓	✓	✓	✓	✓		✓		✓

HF = heart failure, PC = personal computer, BP = blood pressure, ✓ = marks the devices that patients own.

**Table 4 diagnostics-12-00926-t004:** Home technology IHD patients.

Patient	Internet	Smart Phone	Smart TV	Tablet	Laptop	PC	Intelligent Bracelet	BP Monitor	Intelligent Weight Scale
IHD_P01	✓	✓	✓		✓	✓			✓
IHD_P02	✓	✓	✓	✓	✓				
IHD_P03	✓	✓	✓		✓	✓			
IHD_P04	✓	✓	✓	✓	✓	✓	✓	✓	✓
IHD_P05	✓	✓	✓	✓	✓		✓	✓	✓
IHD_P06	✓	✓	✓	✓	✓				✓

HD = ischemic heart disease, PC = personal computer, BP = blood pressure, ✓ = marks the devices that patients own.

## Data Availability

Data available upon request due to ethical restrictions. The data presented in this study are available upon request from the corresponding author. The data are not publicly available due to ethical restrictions.

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
