# Peer review of "Patient Perception When Transitioning from Classic to Remote Assisted Cardiac Rehabilitation"

_diagnostics, 2022, doi:10.3390/diagnostics12040926_

Round 1
Reviewer 1 Report
In my opinion current paper has some POINTS of WEAKNESSES and so, this version is quite poor and I think MAJOR REVISIONS are necessary.
First of all, ENGLISH LANGUAGE requires moderate improvement (i.e. sentences as at lines 21-22 might be totally re-written.
Moreover, quality of tables and charts may be greatly ameliorated.
For enlarging DISCUSSION I'd like to suggest the following articles (also for improvement of references level and besides that CHECK if they are always in accordance with journal's rules for references):
- Telemed J E Health. 2020 Jan;26(1):34-41. doi: 10.1089/tmj.2018.0302.
- Digit Health. 2017 24;3:2055207617705548. doi: 10.1177/2055207617705548.
- Stud Health Technol Inform. 2015;209:9-14.
Finally, I believe not so high priority of publication because of limitations which authors promptly told.
Best regards.
LG
Author Response
Dear Reviewer,
First of all, I would like to thank you for your time and for the extremely helpful suggestions you made to our study. We analysed them point by point and made the changes accordingly.
We hope that the revised manuscript is in line with your expectations.
Kind regards,
Maria Pana
Reviewer 2 Report
Cardiac rehabilitation programs are medical services offered to patients diagnosed with heart diseases.
The authors aimed for patients' perception assessment on the involvement of technology and remote monitoring devices in cardiac recovery.
During a project, the Living Lab phase of the Virtual Coaching Activities For Rehabilitation In Elderly (vCare) project, they evaluated eleven patients (five heart failure patients and six ischemic heart disease patients).
They conducted a system usability survey to assess the patient's perception of the system's technical and medical functions.
I agree with the authors that the survey had excellent preliminary results in the heart failure study group and good results in the ischemic heart disease group.
I is very good that the low access of patients to cardiac rehabilitation in Romania has led to increased interest and motivation in joining this study.
It is a good idea to improve the final version of the survey product to adapt to the patient's percetions.
The article has strong merits however it need major changes before being considered for the acceptance
I suggest the following major cahnges.
- The aim “We conducted a system usability survey to assess the patient's perception of the system's technical and medical functions” is very near to the end of the abstract.
- The introduction is too long (five paragraphs) and the reader could get lost. Please reduce and smoth it.
- The goal does not appear explicitely. Please insert a clear purpose paragraph..
- The writing of the methods is poor. Please rearrange them according to the following parts 1) Participants and procedures; 2) Measures; 3) Statistical analysis;
- Resuts could be rearranged by themes and better developed
- The figures need to be redesigned according to the standards (for example with the bar plots)
Author Response
Dear Reviewer,
First of all, I would like to thank you for your time and for the extremely helpful suggestions you made to our study. We analyzed them point by point and made the changes accordingly.
We have shortened and smooth the introduction as required, with the insertion of a subchapter regarding the study goals.
We have rearranged the writing of the methods as suggested into: Participants and procedures, measures, statistical analysis and results. The Results subchapter has been divided by themes and developed.
The charts have been upgraded as suggested to bar-plots, and the tables have been redesigned.
We hope that the revised manuscript is in line with your expectations.
Kind regards,
Maria Pana
Round 2
Reviewer 1 Report
After revisions I think it is suitable for considering publication.
Best regards.
Reviewer 2 Report
The manuscript has improved.
There are no further comments.